Generalized linear mixed models can detect unimodal species-environment relationships

Jamil Tahira 1 2
ter Braak Cajo J.F. 1 cajo.terbraak@wur.nl
1 Biometris, Wageningen University and Research Centre , Wageningen , The Netherlands
2 Department of Mathematics, COMSATS Institute of Information Technology , Islamabad , Pakistan
Xu Jianhua
Electronic publication date: 2013 Jul 9
Publication date: 2013
Volume: 1
Electronic Location ID: e95
Received 2013 May 6; Accepted 2013 Jun 7
Copyright: © 2013 Jamil and ter Braak
Copyright year: 2013
Copyright holder: Jamil and ter Braak
License: This is an open access article distributed under the terms of the Creative Commons Attribution License, which permits unrestricted use, distribution, and reproduction in any medium, provided the original author and source are credited.
License URL: https://creativecommons.org/licenses/by/3.0/

Keywords: Environmental gradient, Testing unimodal response, Niche theory, Generalized linear mixed model, Gaussian logistic model

Funding: Higher Education Commission Pakistan TJ was funded by a grant from the Higher Education Commission Pakistan. The funders had no role in study design, data collection and analysis, decision to publish, or preparation of the manuscript.

==============================
Niche theory predicts that species occurrence and abundance show non-linear, unimodal relationships with respect to environmental gradients. Unimodal models, such as the Gaussian (logistic) model, are however more difficult to fit to data than linear ones, particularly in a multi-species context in ordination, with trait modulated response and when species phylogeny and species traits must be taken into account. Adding squared terms to a linear model is a possibility but gives uninterpretable parameters.

This paper explains why and when generalized linear mixed models, even without squared terms, can effectively analyse unimodal data and also presents a graphical tool and statistical test to test for unimodal response while fitting just the generalized linear mixed model. The R-code for this is supplied in Supplemental Information 1.

Introduction

Niche theory predicts that species occurrence and abundance show non-linear, unimodal relationships with respect to environmental gradients (Austin, 1987; Palmer & Dixon, 1990; Whittaker, 1967). Many studies fail to test for unimodal response (Austin, 2007). Thus straight-line relationships are often fitted without justification (e.g., Gibson et al., 2004). Pollock, Morris & Vesk (2012) propose a generalized linear mixed model for investigating trait modulation of the environmental response of a number of species. In their data unimodal response was said to be precluded, presumably as they examined relatively short environmental gradients. But might their method or a small modification thereof have worked for unimodal response? Ives & Helmus (2011) recently proposed phylogenetic generalized linear mixed models. Can these models usefully analyze unimodal response?

A similar question arises in community ecological ordination, a class of multivariate methods to analyze the occurrence and/or abundance of a set of species in a set of sites and resulting in a configuration of the sites in a factorial plane, the directions of which can be interpreted as latent environmental variables (Jongman, ter Braak & van Tongeren, 1995; ter Braak & Prentice, 1988; Walker & Jackson, 2011). Principal component analysis and redundancy analysis are linear ordination methods whereas (detrended) correspondence analysis and canonical correspondence analysis are claimed to be able to analyze unimodal response (ter Braak, 1985; ter Braak, 1986). Nevertheless, (canonical) correspondence analysis is an eigen vector method and therefore inherently linear. This is most apparent in the reconstitution formula of (canonical) correspondence (Greenacre, 1984; ter Braak & Verdonschot, 1995). How can it be understood that these methods are able to model unimodal data but are inherently linear?

Some insight in this question is given by Ihm & Van Groenewoud (1984) and further worked out by ter Braak (1987) and de Rooij (2007) who showed the relationship between the unimodal model and a generalized (bi)linear model, also known as Goodman’s RC model. The relationship can be used both ways. Ihm & Van Groenewoud (1984) use the relationship to justify their model B which is a (bi)linear model, for ecological ordination and de Rooij (2007) uses it to transform the linear predictor of the RC model into a quadratic form, with the graphical purpose to transform a vector representation or biplot to a distance representation that is supposed to be easier to interpret for naïve users of multivariate methods. Additional insight is given by Zhu, Hastie & Walther (2005) who introduced a weighted sample model to show the equivalence of canonical correspondence analysis and linear discriminant analysis.

In this paper we propose a graphical tool and statistical test to test for unimodal response while fitting just a generalized linear mixed model (GLMM) without squared terms. GLMMs are model-based, inferential statistical tools. GLMMs are very useful for describing the community patterns and are becoming popular in ecological and evolutionary studies (Bolker et al., 2009; Ives & Helmus, 2011; Zuur et al., 2009). Even when unimodality is detected using the proposed tool and test, we claim that GLMMs can effectively analyze unimodal data when the niche width is not very different among species and illustrate this claim by comparing the GLMM approach with an explicit unimodal model approach on data that show unimodal response. We focus on explicit environmental variables. In this case of direct gradient analysis, we can of course add the squares of environmental variables to the model (ter Braak & Looman, 1986) and test the statistical significance of the addition, but the argument extends to the trait modulated response (Jamil et al., 2012) and to latent variable models (Walker & Jackson, 2011) for which models without squared terms are easier to handle with software that is more widely available.

Material & Methods

Generalized linear mixed models and unimodal response

For ease of exposition we start with a logistic linear mixed model for presence–absence data as example GLMM. The same approach can be followed for count data and loglinear models, which would relate to the RC model (de Rooij, 2007). Consider the logistic linear mixed model that relates the probability of occurrence pij of species j in site i to a quantitative environmental variable xi by the formula (1) logit(pij)=αj+βjxi+γi(i=1,…,n;j=1,…,m)

with αj an intercept, βj a slope and γi a site effect, which we all take as random parameters with normal distributions with zero mean and variances σα2, σβ2 and σγ2. In this random intercept, random slope model (Gelman & Hill, 2007; Zuur et al., 2009) it is prudent to have an additional parameter ρ for the correlation between the intercepts {αj} and slopes {βj}; otherwise the model would change by just centering the environmental variable. Inclusion of the random site effects {γi} are a means to avoid pseudoreplication (Hurlbert, 1984) as they introduce correlation among species. These correlations were not modelled by Pollock, Morris & Vesk (2012) which makes their statistical tests liberal. The site effects may account for the size of the site, the fertility of the site or any other unknown factors that influence the probability of occurrence of all species in the site. The site effect γi will thus be expected to be related to the expected number of species in a site, that is to ∑jpij and, in terms of the data, to the number of species that is observed in a site, for short the site total, defined as Si = ∑jyij. The site total and the site effect are expected to have a monotonic positive relationship.

We now turn to one of the simplest unimodal curves for presence–absence, the Gaussian logistic curve (Oksanen et al., 2001; ter Braak & Looman, 1986) (2) logit(pij)=aj−(xi−uj)22tj2

with aj a coefficient related to maximum probability of occurrence, uj the species optimum and tj the tolerance of species j. This model thus has a logistic form but is nonlinear in this parameterization. By expanding the quadratic term, (3) aj−(xi−uj)22tj2=aj−12tj2xi2−12tj2uj2+1tj2xiuj=aj−12tj2uj2+ujtj2xi−12tj2xi2

it can be fitted to the data of each individual species by a generalized linear model (GLM) by using x and x2 as predictors (Jongman, ter Braak & van Tongeren, 1995; ter Braak & Looman, 1986). The fit is unimodal if the regression coefficient of x2 is negative.

Might we be able to model unimodal response even without the squared term? On assuming tj = t and setting (4) αj=aj−12t2uj2,βj=ujt2andγi=−12t2xi2,

we obtain Eq. (1) again. If t would vary among species then Eq. (1) does not exactly hold because xi2/tj2 then also depends on j. With equal tolerances, unimodal response can thus be represented by a simple linear model with site effects and, as we propose, be fitted by a GLMM based on Eq. (1). The GLMM has additional normality assumptions. In case of unimodal response, the assumption that the site effects {γi} are independent normal is false, as the site effects then depend on xi through xi2/t2. This will be the basis of our test on unimodal response in the next section. The site effects in Eq. (4) also have a nonzero mean, but that is not a problem, as the mean can be taken out and transferred to the intercepts (αj).

The unimodal model with two or more environmental variables (ter Braak & Prentice, 1988) can similarly be rewritten as a simple linear model without squared terms if the tolerances are equal (Appendix 1). In conclusion, up to distributional assumptions, the GLMM (e.g., Eq. (1)) can be interpreted as a Gaussian logistic model with equal tolerances for the species.

A graphical tool and statistical test for unimodal response

Equation (4) suggests a graphical tool for detecting unimodal response and also a statistical test. The idea is to fit a GLMM to the binary data {yij} with respect to the environmental variable with values {xi} (i = 1, …, n). In the R package lme4 (Bates, Maechler & Bolker, 2011) , the model can be fitted by lmer(y ∼ 1 + x + (1 + x∣sp) + (1∣site),family = binomial(link = "logit"),data),

where y represents the vectorized response data while sp and site are factors indicating species and sites. The site effects {γi} obtained from the fit are then plotted against the environmental variable {xi}. There is an indication of unimodal response in terms of the species response with respect to the environmental variable x if this graph shows a n-shaped (as opposed to u-shaped) quadratic relationship. If the shape is not quadratic but curved, a transformation of x may improve it.

In the statistical test for unimodal response, the null model is the GLMM of Eq. (1) with, specifically, independent and normally distributed site effects. The alternative model is that the site effects depend quadratically on the environmental variable. As the site effects typically also depend on the site total S, a sensitive test on unimodal response is obtained by regressing the site effects on x, x2 and S, according to the model formula (5) γ∼x+x2+S.

There is evidence of unimodal response if the squared term is significant as judged by a z-test or, equivalently, an ANOVA test on its regression coefficient, the null model being γ ∼ x + S. The R-code for making the graph and performing the test on unimodality is supplied in Supplemental Information 1.

Simulated data

In the first example series the procedure to simulate data is the following:

(1) Generate n = 50 values of an environmental variable x as a random sample from the uniform distribution, x ∼ U(−2, 2).

(2) Generate a vector u of length m from a uniform distribution U(−τ, τ), where τ = 2 + t, for a fixed value of t, to ensure that optima are also placed outside the sample range of x.

(3) Generate a vector a of length m drawn at random from the standard normal distribution.

(4) Generate binomial probabilities pij from the unimodal response curve (6) pij=logit−1aj−(xi−uj)22tj2

and generate presence–absence data yij at random from a Bernoulli distribution with probability pij and tj = t. We simulate data with constant tolerance in each data set for m = 100 species and vary t between data sets (t = 0.5, 1 and 4). Figure 1 indicates how the simulated species response curves look in each data set. We repeated the example with x and u having normal distributions instead (x ∼ N(0, 1) and u ∼ N(0, t2)), but do not show the results as they were very similar to the uniformly distributed case.

Figure 1 Simulated unimodal response (occurrence probability pij of a species at a site) against environmental variable x for a selection of species.

For left to right: three tolerances (t = 0.5, 1 and 4).

In the second example series, we vary the tolerance lognormally among species, t ∼ Log N(0, σ) with σ = 0.25, 0.5 and 1 so that the median tolerance is 1. For σ = 1, the coefficient of variation is larger than 100%. In the third example series we vary the number of species (m = 10, 50 and 100) with t = 1. For the rest, the simulation is set up as in the first series. We also simulated data according to Eq. (1) with the strict assumptions of the GLMM of independent normal site effects.

Each dataset was characterized by beta diversity and length of gradient. The index of beta diversity is βw = T/S−1, where T is the total number of species, and S is the average number of species per site (Whittaker, 1960). Length of gradient, obtained by analyzing the data with detrended correspondence analysis (DCA), is expressed in standard deviation (SD) units (Hill & Gauch, 1980). Values greater than 4 are commonly taken to indicate unimodal response. Beta diversity was calculated using the asbio package (Aho, 2011) and DCA was performed in the vegan package (Oksanen et al., 2011), both in R.

Real data

We illustrate our method with three real data sets. The first is the Dune Meadow data (Jongman, ter Braak & van Tongeren, 1995). This is a small data set of 28 higher plants in 20 sites in a dune area in the Netherlands. Environmental variables, related to soil and management, were measured at each site; we use the variable Moisture.

The second data set includes the vegetation of the rising seashore on the island Skabbholmen in the Stockholm archipelago, eastern central Sweden (Cramer & Hytteborn, 1987) and is part of the Canoco package (ter Braak & Smilauer, 1998). The data set consists of 63 sites sampled in both 1978 and 1984 and contains 68 species. The environmental variable is Elevation.

The third data set involves phytoplankton communities of 203 lakes located within four climate zones and associated measurements on various environmental variables and morphological species traits of 60 species (Kruk et al., 2011). We consider the environmental variable Temperature.

For each data set we fit a GLMM according to Eq. (1) with x the noted environmental variable, plot the resulting site effects against x and test for unimodal response at the 5% significance level explained by and below Eq. (5). For the seashore data, we analyzed 1978 and 1984 separately. We also compare the regression coefficient βj as estimated by GLMM with the optimum uj as obtained by explicitly fitting Eq. (2) using GLM (ter Braak & Looman, 1986) for species with a well-defined optimum, that is, the squared term of which has a z-ratio smaller than −1 in the GLM model. In the small Dune Meadow data set we used z-ratio < 0.

Results

Simulated data

Figure 1 shows the simulated response curves in example series 1; the sampled range of the environmental variable is the range of x shown. With increasing tolerance the part of the curves that is sampled shows less unimodal response. This is expressed quantitatively in the length of gradient SD units which varies between about 1 SD (not so unimodal) to 6 SD (very unimodal); the beta diversity varies correspondingly between 1 and 5.

In Fig. 2, the site effects estimated by the GLMM analysis of each of the simulated data sets are plotted against the environmental variable. In all three series, site effects shows a clear quadratic relationship with the environmental variable except for large tolerance (t = 4) in the first series. Note the decreasing range of site effects as the tolerance increases in the first row of Fig. 2; for large tolerance, the site effects are close to zero. Nevertheless, the squared term in Eq. (5) was significant in all examples (P < 0.001) so that the method detects unimodal response even if it is moderate (t = 4). When the data are simulated according to Eq. (1) with the strict normality assumptions of the GLMM, the squared term was not judged significant more often than expected on the basis of Type I error of the test.

Figure 2 Simulated data: site effects as estimated by GLMM based on Eq. (1) plotted against environmental variable x.

A quadratic relationship indicates a unimodal response. Rows: example series 1–3, columns: parameter varied.

Figure 3 shows the relationship between the random slopes (βj) estimated by GLMM and the true optima (uj) in the three simulation series. The relation is positive as predicted by Eq. (4). The relationship is weaker the larger the tolerance (Fig. 3, first row). With tolerance varying across species, the relationship continues to hold true surprisingly well (Fig. 3, second row), except perhaps when the coefficient of variation of the tolerance is large (>100%). The larger the number of species the clearer the predicted relationships (last row of Figs. 2 and 3).

Figure 3 Simulated data: GLMM random slopes of Eq. (1) plotted against the true optima of the Gaussian logistic curve of Eq. (2).

Rows: example series 1–3, columns: parameter varied. In the two bottom rows the (median) tolerance is 1.

Real data example

The site effects estimated by GLMM show a quadratic relationship with the noted environmental variable in each of the three data sets (Fig. 4 top row), with the least unimodal response in the Dune Meadow data. Unimodal response is significant in all cases, as judged by our proposed significance test (P < 0.001) and the relationship between the random slopes (βj) estimated by GLMM and the optima (uj) obtained from a fit of the unimodal model of Eq. (1) is close to linear (Fig. 4, bottom row). In the Dune Meadow data there is one outlier for a species with a z-ratio close to 0. In the phytoplankton data, the species with similar low values for the optimum received differential values for the slope, but otherwise there is a good agreement.

Figure 4 Real data: site effects (top) and the GLMM random slopes (bottom) plotted against the environmental variable x and the optima of the Gaussian logistic curve as estimated by GLM, respectively.

Discussion

Our analysis of multi-species data sets showed that a GLMM without squared terms but with site effects is able to detect unimodal response. The theory required equal tolerances among species, but the simulations showed remarkable robustness to this assumption. With significant unimodal response, as judged by our test, the assumption of independence and normality of the site effects underlying the GLMM is clearly violated. This result can be interpreted in two ways. The first way is to try and adapt the model so that the assumptions are no longer (grossly) violated, for example, by extending the GLMM with an explicit square of the environmental variable as fixed effect, yielding (7) logit(pij)=αj+β1jxi+β2xi2+γi

and then testing the normality assumption on the new site effects, for example by making a Q–Q plot (Gelman & Hill, 2007). The site effects γi obtained from the fit can then be plotted against the environmental variable to assess whether or not the unimodality is adequately modeled using quadratic terms. The extended model still assumes constant tolerance (as does Eq. (1)) as it can be rewritten as (8) logit(pij)=aj−(xi−uj)22t2+γ˜i

with (9) t=1/−2β2,uj=t2βj,aj=αj+12t2uj2,andγ˜i=γi+12t2xi2.

We can go one step further and test the assumption of equi-tolerance by adding the squared term x2 also as a random (species-dependent) component to Eq. (7) and testing the significance of this extra variance component. In the R package lme4 (Bates, Maechler & Bolker, 2011), the two models to compare are (with xx = x2) lmer(y ∼ 1 + x + xx + (1 + x∣sp) + (1∣site)family = binomial(link = "logit"),data)

and lmer(y ∼ 1 + x + xx + (1 + x + xx∣sp) + (1∣site)family = binomial(link = "logit"),data).

We are currently investigating robust ways of testing the equi-tolerance assumption on the basis of these two models, as the default ANOVA test has inflated type I error for small t.

The second way of interpretation is to conclude that the GLMM based on Eq. (1) is remarkably robust to the normality assumption on the site effects and the equi-tolerance assumption and that it can be used as a basis of more complicated models, such as the trait modulated response model (Jamil et al., 2012; Pollock, Morris & Vesk, 2012) and latent variable models (Walker & Jackson, 2011), even for unimodal response. In this way, the site effects are treated as nuisance parameters and the independence and normal distributions, needed for efficient computation, as a prior distribution that a posteriori may turn out to be false. Cormont et al. (2011) used this type of rationale to claim that their linear trait-environment method is well suited to analyze unimodal response.

Conclusions

Site effects in multi-species GLMM serve three purposes in ecological data, first to avoid pseudoreplication (Hurlbert, 1984), second to account for differences in species richness among sites (Ives & Helmus, 2011) and third (this paper) to allow for common nonlinear, unimodal response. The results of this paper imply that the phylogenetic generalized linear mixed models of Ives & Helmus (2011) are not in conflict with niche theory as they include random site effects and can thus deal with unimodal response.

Walker & Jackson (2011) used a latent variable approach to test for unimodal response. We tried their approach with the phytoplankton data, but failed to get an answer because the program for fitting the quadratic model crashed. We focused this paper on the easier task of detection unimodal response to measured environmental variables by using a GLMM without squared terms. To scale up to a latent variable approach we need factor analytic structure within a GLMM. This already exists for linear mixed models (Thompson et al., 2003; Verbyla et al., 2003) and it is a matter of time that it becomes standardly available for GLMM. This paper shows the utility of such factor analytic models in ecology (Walker & Jackson, 2011) if they allow additional random site effects.

A GLMM with terms that are linear in quantitative predictors is, of course, linear. But with random site effects included, GLMMs can detect and fit unimodal response, with the provision that the differences in niche widths among species is not too large. The application scope of GLMM in ecology is thus much broader than one might think at first glance.

Supplemental Information

Supplemental Information 1 R-script for testing unimodality with output and plot for the dune meadow data.

The zip file contains three files. The file “Rcode_with_example.r” is the R-script with R-function Test.Graph.unimodal and application to the dune data. The file “Rcode_with_example_output.txt” contains the output of the R-script and the file “Rcode_with_example_plot.pdf” the produced plot.

Click here for additional data file.

This paper benefited from input from Mike Palmer, Steve Walker and Mu Zhu.

Appendix 1

This appendix shows that similarity between the Gaussian logit models and GLMMs extends beyond the Gaussian logistic curve. The Gaussian logistic model can be extended to two environmental variables as follows (ter Braak & Prentice, 1988) logit(pij)=aj−12(d1(xi1−u1j)2+d2(xi2−u2j)2−2d12j(xi1−u1j)(xi2−u2j))

where d’s are precision parameters, in the context of the bivariate normal distribution (Rue & Held, 2005). By setting αj=aj−12(d1u1j2+d2u2j2−2d12ju1ju2j),

β1j=d1u1j−d12ju2j,

β2j=d2u2j−d12ju1j,

β3j=d12j,

and γi=−12(d1xi12+d2xi22)

we can write logit(pij)=αj+β1jxi1+β2jxi2+β3jxi1xi2+γi.

Here β3j are random effects for interactions. If the ‘co-precisions’ are equal (d12j = d12) the term β3jxi1xi2 can be subsumed in to the site effects γi and the model can do without interactions. The γi account for the quadratic term arising from the Gaussian (logit) model. Extension to more than two environmental variables is immediate. In conclusion, up to distributional assumptions, the GLMM can be interpreted as a Gaussian logistic model with equal tolerances for the species.

Additional Information and Declarations

Competing Interests

Author Contributions

CJFtB is an Academic Editor for PeerJ.

Tahira Jamil conceived and designed the experiments, performed the experiments, analyzed the data, contributed reagents/materials/analysis tools, wrote the paper.

Cajo J.F. ter Braak conceived and designed the experiments, contributed reagents/materials/analysis tools, wrote the paper.

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
