# Peer review of "Generalized linear mixed models can detect unimodal species-environment relationships"

_PeerJ, doi:10.7717/peerj.95_

## Round 0.1 · original submission · Minor Revisions

Dear Dr. Tahira T Jamil,
I have received the the review reports on your manuscript, "Generalized Linear Mixed Models can detect unimodal species-environment relationships", which you submitted to PeerJ.
Based on the reports, your manuscript can be accepted for publication after minor revisions. When preparing your revised manuscript, you are asked to carefully consider the reviewer comments, and submit a list of responses to the comments.

·

Basic reporting

There were some English issues, which I am fairly certain were typos. Otherwise the article is well-written.

Experimental design

Not relevant given the methodological / theoretical focus of the study.

Validity of the findings

The conclusions reached by the authors follow from the results presented.

Additional comments

General comments
* * *
This manuscript argues that unimodal species response curves can be estimated using models that contain a random site effect, instead of the more direct approach of including quadratic terms into the model. A simple diagnostic for detecting unimodality is also presented and tested using simulations.

An honest comparison of the authors' random site effect approach with the explicit quadratic term approach is given. I appreciate this honest discussion. As the authors point out, the random site effect approach requires the assumption of equal tolerances among species (lines 235-6) and that their assumption of independence and normality of the random site effects will also typically be broken in data sets with unmodality (lines 236-8). However, they show that their method is robust to these violated assumptions in a variety of ways and point out that their method has computational advantages that could be very useful in applied problems (particularly in latent variable modelling).

Overall this manuscript provides a useful alternative approach to modelling unimodal species response curves and provides a balanced discussion of its potential. This manuscript represents a very useful contribution.


Specific (mostly minor) comments
* * *
line 35: 'models' not 'model'.

line 51: Something is wrong with parentheses here.

line 87: "they would have been a welcome addition…" is maybe a little too informal in my opinion. How about "these correlations were not modelled by Pollock et al. (2011), which may have caused model specification problems"? Even that is still vague, but I think its better.

line 91: "in terms to the data" should probably be "in terms of the data".

line 93: The word 'naturally' is vague here. I know that this term is common in the math/stats literature, but if ecologists are the desired audience they may be confused. Why not say "The site total and the site effect are expected to have a monotonic positive relationship"?

line 119: I really like this sentence. It nicely states a simple and useful result. Maybe it would help to add the equation number (e.g. Eq. 1) after "GLMM" to be more specific?

line 136: 'quadratically' not 'quadratic'.

lines 140-1: I have several thoughts about these two lines:
(A) Am I right in thinking that the degree of scatter around equation (5) gives an indication of the degree to which species have equal tolerances? Would zero residuals indicate estimated tolerances that are exactly homogeneous among species?
(B) I'm wondering about the statistical theory here. Isn't there a circularity problem with the t-test assumptions given that the response, \gamma, is estimated from a model that includes the predictors, x and S? I would feel safer with a permutation test or a bootstrap. Although maybe it doesn't matter in this case? See my comment below for line 261 on error rates.
(C) Might it help to relate equation (5) to the idea of an 'arch effect'?

line 148: I don't like how the symbol 't' has a different meaning from the tolerance. A different symbol might be less confusing.

lines 236-238: Excellent point.

line 239: 'assumptions' not 'assumption'.

lines 239-261: I have to admit that this is the kind of interpretation/approach that I would take, even though the authors prefer their second interpretation. However, I admit that my preference is largely a matter of taste and I welcome the alternative provided here.

lines 261: 'models' not 'model'.

line 261: Have the error rates been checked for the p-values obtained using the approach of equation (4)? If not, then this criticism of the "default ANOVA test" is a little unfair.

lines 261-264: This is an interesting robustness claim. I think that a productive line of future research could be to explore the limits of this robustness.

·

Basic reporting

The paper describes the rather surprising result that parameter fitting for unimodal gaussian curves can be linearized if the tolerances (standard deviations of the curves) are assumed to be equal. The method is demonstrated with some simulations and three real datasets. The submission has a clear introduction and is self-contained. In the abstract, I did not understand what was meant by saying that adding squared terms "gives uninterpretable parameters". Line 349 - reference to Pollock et al. is incomplete.

Experimental design

The methods are mathematical and are set out clearly. There is no overall experimental design as the authors aim for proof of concept.

Validity of the findings

The conclusions are somewhat speculative, indicating how the method might be used in future. The authors clearly show that the method suggests a unimodal response in the examples given.

---

## Round 0.2 · accepted · Accept

I am pleased to inform you that your manuscript has now been accepted for publication. Thank you again for submitting your manuscript to The PeerJ.